# Alcohol, Other Drugs Use and Mental Health among African Migrant Youths in South Australia

**DOI:** 10.3390/ijerph18041534

**Published:** 2021-02-05

**Authors:** Lillian Mwanri, William Mude

**Affiliations:** 1College of Medicine and Public Health, Flinders University, Adelaide 5042, Australia; 2School of Health, Medical and Applied Sciences, Central Queensland University, Sydney 2000, Australia; w.mude@cqu.edu.au

**Keywords:** African migrant and refugee youths, mental health, alcohol and other drugs, integration, South Australia

## Abstract

This paper was part of a large study that explored suicide among African youths in South Australia. The paper reports perspectives about alcohol and other drugs (AOD) use and mental health among African migrant and refugee youths in South Australia. The study employed a qualitative inquiry, conducting 23 individual interviews and one focus group discussion with eight participants. An acculturative stress model informed data analysis, interpretation and the discussion of the findings that form the current paper. African migrant and refugee youths revealed challenging stressors, including related to cultural, socioeconomic, living conditions, and pre- and post-migration factors that contributed to mental health problems and the use of AOD in their new country. The traumatic loss of family members and social disruption experienced in their countries of origin were expressed as part of factors leading to migration to Australia. While in Australia, African migrant and refugee youths experienced substantial stressors related to inadequate socioeconomic and cultural support, discrimination, poverty, and unemployment. Participants believed that differences in cultural perspectives about AOD use that existed in Africa and Australia also shaped the experiences of social stressors. Additionally, participants believed that these cultural differences and the identified stressors determined AOD use and mental health problems. The findings highlight the need to understand these social and cultural contexts to improve mental health services and help reduce the use of AOD, which, when problematic, can influence the health and integration experiences of these populations.

## 1. Introduction

### 1.1. Migration and Resettlement of African Refugees

Across the world, the ease of travel and international mobility have increased, including for people from Africa. Additionally, in 2000, the United Nations High Commissioner for Refugees (UNHCR) called for the need to provide resettlement opportunities for refugees who have spent several years in refugee camps in challenging conditions [1], especially from the African continent. Following the UNHCR calls, the Australian Government declared a humanitarian commitment to resettle refugees from African nations, particularly those living in protracted refugee situations from the Horn of Africa. This commitment resulted in a sharp increase in the proportion of refugees resettled from African countries, rising from 33 per cent in 2003 to 70 per cent in 2005 [2]. In 2012–2013, over half of visas granted under the Humanitarian Program were allocated to people born in Sub-Saharan Africa, North Africa, and the Middle-East, with 39.2 per cent of all persons granted visas being aged between 0 and 17 years [3]. There were 317,182 people born in Sub-Saharan Africa in Australia in 2016, and over 20,000 were living in South Australia [4].

During the process of immigration, migrant and humanitarian refugees are exposed to multiple stressors [5,6]. Studies among African migrants (the majority of whom have a refugee background in Australia) have reported significant stressors relating to inadequate employment, housing, education, and integration following settlement [7,8,9]. Additionally, there are considerable and diverse migration trajectories experienced between and within refugee groups. Pre-migration contexts such as cultural backgrounds, the country of origin and circumstances surrounding the decision to immigrate can affect groups and individuals in a range of ways, including poor mental health outcomes [10].

### 1.2. Migrants and Refugee Youths, Alcohol and Other Drug Use and Mental Health

The age at which a migrant and refugee migrate, and the related settlement opportunities and challenges can have a profound influence on mental health. A study suggests that compared to other migrant groups, migrant youths are more likely to be at a higher risk of suicide [11]. Migrant and refugee youths experience an increased risk of self-harm behaviors and are vulnerable to suicidal ideation because of challenges to the social factors following resettlement, described previously [12]. Migrant and refugee youths are also at risk of increased alcohol and other drugs (AOD) use, which increases individual risks to mental health problems and suicide [13]. AOD use among migrant and refugee youths has also been linked with experiences of social, emotional, and behavioral problems, including feeling depressed [14].

In Australia, these are serious issues because a large proportion of young people who emigrated from war-torn areas in Africa and the Middle East in the last decade arrived unaccompanied, without parents or guardians, in their critical stage in life [3]. It is important to point out that despite the increased risks of migrant and refugee youths to AOD use and mental health, African migrant and refugee youths are resilient, which play protective roles in reducing the use of AOD and related harms [15].

Although previous research has explored the resettlement and integration particularly of African refugees [16], there is a dearth of research examining the beliefs about AOD use and mental health among African migrant and refugee youths in Australia [17]. African migrant and refugees, including youths, are under-serviced by AOD and mental health services. This inequity in service requires more research to enhance an understanding of this community [18].

### 1.3. Aims of the Study

This study was commissioned by the African Communities Council of South Australia (ACCSA), an overarching organization for African communities in South Australia, to determine the perspectives of African youths (the majority of whom has a refugee background) in South Australia on suicide, AOD use and mental health following increased suicide among youths in this community. This paper presents part of the findings from this project and explores the perspective of African migrant and refugee youths on AOD use and issues of mental health. This study aims to contribute to the limited existing literature on AOD use and mental health problems in the African communities by understanding the contexts in which AOD use and mental health issue occur among African migrant and refugee youths in Australia. We attempted to achieve this aim by answering the research question: What are the main factors that influence AOD use among African youths in South Australia?

## 2. Methods

### 2.1. Theoretical Framework

We employed the acculturative stress model to guide our data analysis and interpretation. This model focusses on understanding the unique stressors that are rooted in the process of acculturation [19]. Acculturation is a process of cultural change experienced by migrants following a contact with or living in a different cultural environment, and it is understood to be complex and dynamic [20]. According to Berry, Kim, Minde, and Mok [19], the process of acculturation and related stressors can lead to a stressful experience of resettlement and integration. They argue that individuals might have different experiences during the process of acculturation and that stressors might depend on the degree of their experience.

Culture and identity, reasons for migration, language, demographic characteristics, social practices, and cultural values are reported factors moderating experiences of acculturation and stressors [21]. Acculturative stress is reported to be a risk factor to mental health outcomes among refugees [22]. Although the concept of acculturative stress model has been used in studying different refugee groups, the use of this model is needed to understand the nature of stressors resulting from acculturation of African refugees, particularly youths following their resettlement in Australia.

### 2.2. Study Design and Recruitment of Participants

The study employed a qualitative inquiry and respectively conducted 23 face to face interviews and one focus group discussion with eight youths aged 18–25 years who lived in South Australia. Participants had arrived in South Australia between 2000 and 2012. Participants were purposively selected to ensure we had representation from different age groups, year of arrival, and countries of origin from Central, Eastern, and Western Africa. Although the majority of the participants had a refugee background, participants who did not have a refugee background were also included recognizing that the youth population is susceptible to a mental health issue in their new country [6]. Because we have included both migrant and refugee youths, we will from here on use African migrant youths to denote both participants with or without a refugee background and we acknowledge the overrepresentation of those with a refugee background among the study participants.

We recruited participants through an African community youth worker who invited potential youths who spoke conversational English to enroll and participate in the study. This method of recruitment was a strength for the project in that, it did draw from the perspective of the community strength, using the youth worker’s position, and not rely upon other service providers to link researchers to participants. The researchers explained the aim and scope of the research to the community youth worker who then invited other youths to participate. Individuals then referred the researchers on to friends who were eligible to be invited to participate face-to-face. All participants were informed about the study and accepted to participate voluntarily.

Data collection included 23 one-on-one interviews and one focus group discussion with eight participants. All interviews and focus group discussion were conducted in English and held at a place where young people felt comfortable.

The interview and focus group guides (Table 1) explored issues on premigration experiences, challenges and difficulties faced in Australia, why some young people may use AOD or take their own life, how they come to use drugs or alcohol, the impacts of AOD use, how they would describe the AOD use among their friends, issues concerning AOD and mental health, social networks and support, family relationships, seeking support from mental health and AOD services, and community support. The interview and focus group discussion guides were informed by research literature and questions were designed to ensure flexibility to expand on points of interest and to explore issues that were important for the topic [23]. Participants were encouraged to actively engage in expressing their views and were made to feel safe talking about stigmatized issues.

### 2.3. Ethical Considerations

The Flinders University Social and Behavioral Ethics Committee (SBREC) approved the study protocols. The permission to collect information from members of the community was also obtained from ACCSA. Each study participant received an information sheet outlining details of the research and its purpose before commencing interviewing or focus group discussions. Participants were assured that their identity would be kept confidential and all names would be replaced with pseudonyms. Although they all lived across the suburbs of Adelaide, some, but not all focus group participants knew each other.

Researchers reminded participants that information disclosed in the focus group discussion needed to remain confidential. All participants provided written consent and received information on their rights to terminate their participation at any time during the focus group discussion or the interview. Each participant received a list of professional counselling agencies and offered a free, private, and anonymous follow-up counselling session with costs covered by ACCSA. Each participant was reimbursed $30 for their time and other expenses incurred to participate in the study.

### 2.4. Data Analysis

The focus group discussions and interviews were audio-recorded and transcribed verbatim. Both authors are experienced qualitative researchers, were part of the investigating team, and analyzed the transcripts to ensure credibility, rigor, transparency, and validity of the analytic process [24].

To provide coherence and structure [25], we used the framework approach described by Ritchie and Spencer [26]. The framework identified five steps of data analysis involving familiarization with the data, coding and identifying themes, indexing, charting, mapping, and interpreting the data. In line with this framework, the researchers identified passages of text according to the context, coded them to several relevant categories and assigned the related codes into themes. The analysis was dualistic including inductive, with categories emerging purely from the data and deductive, with categories derived from prior knowledge and the thematic approach enhanced the rigor, transparency, and validity of the analytic process [24,25]. Table 2 provides a snapshot of data analysis for the current paper.

The research team members are experienced in working with young people from African backgrounds and were also members of the African community in South Australia. This background was vital across the continuum of the project and eased the data collection, analysis, and interpretations. Additionally, being members of the African community facilitated trust and rapports between the participants and researchers, although it could have played out exactly in the opposite way.

## 3. Findings

### 3.1. Characteristics of the Study Participants

The average age of the interview (Table 3) and focus group (Table 4) participants was 22 years old. The interview participants involved 8 females and 15 males from five different countries, including the Democratic Republic of Congo (DRC), South Sudan, Liberia, Ethiopia, and Burundi. The focus group members consisted of six females and one male originally from South Sudan, Ethiopia, Ghana, Liberia, Burundi, and Somalia. Table 3 and Table 4 provide the characteristics of the study participants.

There were three themes describing the perspectives of African migrant youths in South Australia reported, including (i) Pre-migration contexts and losses, (ii) Post-migration contexts and realities, and (iii) Contextualizing AOD use and mental health. These themes are described in detail below.

### 3.2. Pre-Migration Contexts and Losses

The narratives from the participants revealed that most African youths arrived in Australia as refugees and stressors related to separations from or witnessing deaths of family and relatives while fleeing from home countries had significant life impacts. The pre-migration deaths of parents meant that many youths came to Australia with relatives or sole parents. Some youths described their relationships with their guardians as strained as the below quote demonstrates:


*I don’t have parents. My mum passed away. And most of these young people here they don’t have like family and some of them might have a mum, but they don’t have a dad. And they might struggle with the family, but most of them don’t talk about it.*
(Participant 21)

Also, some youths fled their countries of origin and lived for many years in refugee camps in the countries of asylum. Some had to move from one country of asylum to another in challenging circumstances as the following quote encapsulates.


*All my parents and the other families stayed in South Sudan. So, we first went to Uganda, we stayed there for a couple of months, in the northern part of Uganda. At the time, it wasn’t safe; there was no difference between Northern Uganda and South Sudan because of the war in South Sudan and the Lord Resistance Army operating in Northern Uganda. We stayed there for a couple of months, but every night was a nightmare. We moved from Northern Uganda, and we went to Kenya, and we arrived in Kenya in 1995. So, we stayed in the refugee camp from 1995 until 2003 when we finally came to Australia.*
(Participant 14)

Similarly, respondents described the experiences of separations from families and uncertainties related to not knowing whether their parents were alive or dead, factors that had a lasting emotional impact and effects on their growth and resettlement in a new country.


*Before I came to Australia, I didn’t know that I had a family. When I came to Australia, people started calling me up and saying, ‘we are your parents’, and I just said, ‘oh my God’. I didn’t know my family because during the war you don’t live with your parents.*
(Participant 20)

Furthermore, as a result of frequent movements from one country of asylum to another, and before arriving in Australia, African migrant youths reported to have lost important opportunities, including educational disruption. Some youths noted that these disruptions were a hindrance to continuing their education in Australia and were forced to grow up early to meet social obligations including looking after younger relatives and marrying at a young age. These experiences are illustrated in the following assertion:


*Because of the war, I was in a camp. I started my education there from year three to year 12. When I came to Australia, I went to the factory, I work. I decided to marry, so I went back home and proposed my wife to come to Australia. My wife came with three boys, one is my nephew, and two are cousins. So, from there, I decided not to go back to school because it has been a long time plus I have personal issues.*
(Participant 22)

### 3.3. Post-Migration Contexts and Realities

Upon resettlement in Australia, some participants experienced poverty, poor employment outcomes, unfamiliarity with the new environment, and complex systems and challenges, including language and cultural barriers.


*Coming to Australia was a sudden thing, like ‘oh you’re coming to this new place’. Like I was young and so didn’t know how it was.*
(Participant 16)


*I know a few people even now they don’t have Centrelink [social security assistance in Australia] money and they don’t have an income. They just live with their parents, but the parents are poor too, they don’t give them an income. They are looking for work, but they can’t get a job, so they have to survive like that.*
(Participant 20)

As a result of disadvantages including poverty, low English proficiency and the difficulties associated with growing up as a young person without parents, these youths were desperate and found it difficult to access job opportunities, ending up with poor resettlement and health outcomes.


*Young people in our community, they have no access to jobs—and because they don’t have access to jobs, they find something to keep themselves busy. Also, because we come from Africa, some of us don’t know English. Some of us, they came here they don’t have parents—their parents died—and they come when they are single… like the cause of all the bad thing is because you don’t have a good quality of life, and if you don’t have a good quality of life, you think you can do bad things.*
(Participant 5)


*They’ll be like saying they’re looking for work, but they’re not going to get a job because of their low level of skill. They can’t get a job, and they’ll be like ‘what can we do?’*
(Participant 20)

Participants reported being anxious, bored, depressed, and perceived themselves as having failed to improve their lives in their perceived land of opportunities, which they reported led youths to antisocial and destructive behaviors, including substance use as a coping mechanism. The following statements encapsulate these claims:


*It goes back to the main problems like depression, anxiety, all these kinds of things. When you start to experience those kinds of things, then you tend to put your problems into alcohol to forget the problems.*
(Participant 14)


*When they get depressed what I see a lot and is more common, is they tend to drink alcohol a lot. They tend to drink and pretty much do things like maybe smoking weed and all that stuff.*
(Participant 21)

### 3.4. Contextualising AOD Use and Mental Health

There was a common consensus among the participants that African migrant youths’ experiences of loneliness, the loss of informal social networks and family ties in addition to their circumstances in Australia made them vulnerable to social peer pressure, which seemed to facilitate AOD use. One participant observed, “*sometimes peer group. Peer group is one of them, I think. If you fall into the wrong group and the environment*” (Participant 6). Other participants also expressed a similar view.


*When you have friends that take alcohol or other drugs, so you tend to follow so that if you want to fit into that friendship or that company, you have to do what they do.’*
(Participant 15)


*I have some of my friends here, they use drugs, they ask me … they give me—like smoking or drinking.*
(Participant 13)

A common agreement among the participant was around the view that there are high expectations on youths in their community, and they are expected to be respectful to maintain their ‘reputation’. However, this high expectation sometimes led to a breakdown of personal and social relationships with the broader community, an issue identified by participants as perpetuating AOD use and mental health problems. Additionally, there was a view among the participants that any ‘reputational damage’ in the community is hard to mend, leading to youths feeling trapped and marginalized from their community.


*If you become frustrated, maybe people don’t pay respect to you sometimes. You say there is nothing to fear again because I already lost that reputation’. That’s the most important thing in my community, is if you lose that reputation in the community it’s very hard to get it back.*
(Participant 15)

Similarly, respondents noted that the breakdown of intimate relationships influenced substance use among youths, a view demonstrated by the following statement.


*They come to drug and drinking alcohol because sometimes their relationships break up. They start smoking, and they start drinking, and they start using drugs. And they think doing these things can help them.*
(Participant 13)

Complex and conflicting cultural and contextual interplays between the Australian and African communities encouraged the use of AOD among African migrant youths in Australia. Unlike in Africa, the Australian society did not prohibit excessive alcohol use, a freedom which participants revealed African migrant youths preferred but created tensions with parents. The lack of social consequences in Australia and access to government financial safety nets encouraged the use of alcohol, as illustrated here.


*Well because it’s a new environment whereby if you have the money you can buy whatever you have and there’s the freedom—so the parents are going to advise their kids, but they do not listen to their parents.*
(Participant 3)


*There is alcohol in Africa, but there’s a way if you’re going to drink alcohol—and you know there’s no Centrelink—if you’re going to drink alcohol you’re going to die because you’re not going to have any other help.*
(Participant 7)

Participants noted the negative impacts of AOD use on physical, psychological, and social health, with risky alcohol consumption both arising from and perpetuating stressors related to acculturation amongst African youths. The following statement demonstrates this sentiment among participants.


*Well, I think alcohol is a health issue because a lot of young people are drinking alcohol. The way I see it, drinking is not the problem, but it’s the amount of alcohol they drink at one time is the problem because if you drink too much, then it becomes heavy on you.*
(Participant 14)

There was acknowledgement among participants that AOD use leads to antisocial behavior and poor decision making. Participants identified poor decision making typically presenting as violence and inappropriate sexual behaviors, and young youths are being vulnerable to offences committed by older youths within their social group when intoxicated.


*Underage drinking has become a problem because a lot of young boys, a lot of young girls under the age of 18 are drinking heavily. And as a result of that, they end up doing a lot of silly things like sexual offences. So they expose themselves into sex earlier because of alcohol and drugs. Young people, as young as 15 and 16, you know, just because they drink and take drugs too they become vulnerable to older people that hang out with them.*
(Participant 14)

Several participants also elaborated on this sentiment with one respondent suggesting that a loss of control with intoxication represented a manifestation of the inability to control turbulent internal emotions. The result of this was violence and arguments with others.


*I think that [alcohol] is dangerous. It leads to a lot of disasters because it makes—teenagers especially when they have alcohol they lose control, they don’t know how to handle that or control their emotions and stuff like that. They start arguing, they start fighting and stuff like that, so I think it’s a big problem for teenagers.*
(Participant 8)

Participants recognized alcohol as an accomplice to suicide, although they did not clarify the precise role that alcohol plays in suicide ideation, attempts, or completion. Respondents observed that alcohol consumption precipitates suicide ideation in youths with underlying psychological problems as the following comment demonstrates.


*If their mind is not working, like sometimes if you take alcohol, you know that cause actually somebody to commit suicide.*
(Participant 6)


*You know mostly these young kids, they tend to drink and with other people making troubles. They drink, and then they try to kill each other and make themselves suicide because of those factors.*
(Participant 5)

## 4. Discussion

This study explored the perspectives of African migrant youths in Australia about AOD use and mental health issues. Acculturative stress model provided the framework to understand these issues in migrant youths because of scarce internal and external coping resources available to them when adjusting to life in Australia [27]. Upon numerous challenges that face the youths in the current study, participants were negotiating not only the transition to a new culture but also that of meeting the expectations of their community. It has to be understood that, as a result of losses including of parents, some participants with refugee backgrounds arrived in Australia as unaccompanied minors, had extended stays in refugee camps with unmet basic nutritional, educational, or recreational needs, contributing to a poor start of new life in Australia. A combination of social and peer pressure, vulnerability, and complexities of their social deprivation could influence substance use, ultimately leading to poor social and health outcomes [28].

An important aspect of the acculturative stress model is how stressors experienced because of a lack of internal and external coping resources during acculturation contribute to stresses [27]. While the paper emphasizes the role of acculturative stress, it is important to acknowledge that there are also other factors that could contribute to AOD use and mental health issues among the participants. For example, pre-migration experiences such as traumas of past lives and losses could contribute to mental health issues among youths and the use of AOD as coping mechanisms. The current study shows that participants experienced loss, separation from families and friends, and uncertainties for their future before and after migration. For example, settlement issues such as lack of employment and underemployment could be additional factors that would lead to AOD use. There are views from the data which suggest that such experiences resulted into extreme stresses, which supports previous evidence that young people with refugee backgrounds experience extreme challenges predisposing them to a wide range of poor health and social outcomes [29]. Participants in this study revealed being trapped in vicious cycles of social disadvantages and socioeconomic situations that affect individuals’ health outcomes, particularly mental health [30]. The data from this study show that unemployment was a common issue facing African migrants youths in Australia, which contribute to their impoverishment and marginalization, often in a state of poor physical and mental health, leading to them using AOD as a coping mechanism. These findings support the link between a lack of employment opportunities and poverty as significant determinants of health [31].

Additionally, missed opportunities and the lack of education due to circumstances described elsewhere in this paper meant that African migrant youths felt excluded and at times discriminated against, for example, when accessing social security support. Discrimination is a known stressor and social determinants of health which hampers the socioeconomic opportunities and has implications for AOD use, poor social outcomes, and mental health [32,33]. Moreover, participants in the current study also revealed living in poverty and experiencing other deprivations. These complex experiences, coupled with AOD use and past trauma could shape the way individuals experience mental health issues in this population group [10]. Alternatively, the poor socioeconomic, harsh environment, and other disadvantages and mental health issues experienced by participants in this study because of past trauma could contribute to this population group having a unique experience of AOD use [34,35].

Consistent with other studies that have demonstrated the influence of alcohol on antisocial, disruptive behaviors [36], and poor health among young people [35], the views expressed by participants in this study suggest that AOD use may contribute to many social, emotional, and behavioral problems among African migrant youths. Participants in the current study revealed that the use of alcohol could contribute to risky behaviors and overall vulnerability of youths. Additionally, the participants in this study acknowledged that many migrant youths use alcohol regularly, and this is concerning because the early onset of alcohol use can contribute to alcohol-related behavioral problems later in adolescence or adulthood [37,38]. Earlier research found that people who began drinking before age 15 were four-fold more likely to develop alcohol dependence during their lifetime than were people who started drinking after 21 years old [39].

In Australia, drinking alcohol is a common characteristic of social life among youths and perceived as a pleasant social norm [40]. For African youths in Australia, it is reasonable to argue that drinking is part of assimilation and acculturation processes [41]. Young people adopting to AOD use can be perceived as embracing the norms in their new society [28]. It is also likely that alcohol use among African migrant youths is rising because of conflicting attitudes between African and Australian cultures regarding their use. In Africa, alcohol use by youths is restricted and often carry severe consequences, whereas it is socially accepted in Australia [40]. Previous studies have supported the assimilation and acculturation process as facilitators shaping the behaviors of migrants [28,42].

To this end, it is worth acknowledging that African youths are resilient but also vulnerable because of the social settings and circumstances in which they live. The results in the current study seem to justify the need for broader social determinants of health approach and provide opportunities for service providers to work with youths in this population to improve AOD and mental health services. These youths can be empowered by service providers through targeted initiatives to become ambassadors and agents of change in their communities. The information from this study has influenced youth-specific preventive mental health interventions and programs run by the African Communities Council of South Australia.

### Limitations and Strengths

One limitation worth noting is that the data were collected eight years ago. However, the data are still relevant because similar issues exist in our community. As such, the need to advocate for programs and policy on AOD use and mental health issue for migrant youths in this community persists. The paper has the potential to contribute to the scant body of literature on AOD use and mental health issue among African migrant youths in Australia. A lot of the views expressed by participants in the current study were about their peers and provided essential insights into AOD use and mental health issue in this population. Another limitation was the use of English language in conducting focus group discussions and interviews, although only participants with conversational English were enrolled. Therefore, this limits the transferability of the findings to youths who do not speak English as they may have different experiences or barriers that need to be addressed. Participants of this study were recruited from metropolitan Adelaide and might have missed African youths living outside of Adelaide. However, the use of an African youth worker to engage participants in the study was a strength, facilitating capacity building in the community. Additionally, the strength of the evidence provided by the findings of this study outweighs its limitations and thence adding to the body of knowledge. Moreover, the researchers’ knowledge and lived experiences of the African culture and context in Australia has shaped the data analysis and interpretations by focusing on pertinent issues that require attention to support African migrant youths in Australia.

## 5. Conclusions

African migrant youths in Australia revealed significant challenges and barriers which were rooted in their pre-and and post-migration contexts. As a result of needing to adapt to their new society, there was a consensus among the study participants that African migrant youths were conflicted, and their effort to fit in led to adverse outcomes, including AOD use and mental health issue. The findings of this study are important and provide some insight into the contexts in which AOD use and mental health issue occur among African youths in Australia. Recognizing these contexts calls for interventions that address the underlying social determinants of health among African migrant youths in Australia. Addressing these socio-environmental factors that foster distress including employment opportunities and culturally unsafe service provision is necessary in order to provide effective and supportive services for these population groups and improve their mental health and wellbeing.

## Figures and Tables

**Table 1 ijerph-18-01534-t001:** Focus group discussion and interview guides.

Tell me about your community in South Australia (where you have come from, social life and social networks and support etc). How has it been like for you here in Australia? Challenges/successes? What problems/challenges/issues do you think young African people face in Australia? What would you say are the main health and social issues that affect young people in your community in South Australia? (if not mentioned, ask: what are your views about alcohol use and mental health issue among youths in African community?) Why are these issues affecting youths in your community? How are these issues different from when youths were in Africa? How do youths cope with these challenges? What do you think the community should do to help African youths? There have been cases in recent times some young people took their own lives in the African community; why do you think young people in the African community take their own lives?

**Table 2 ijerph-18-01534-t002:** Examples of thematic analysis and interpretation of study data.

Extracted Texts	Codes	Themes
- Parents have died - I didn’t know I had family - Living without parents	Separations and fragmentations of families	Pre-migration contexts and losses
- Difficult life - Studying in refugee camps - Restricted in refugee camps	Lack of opportunities and access to jobs	
- Depending on rations - Hard life - Living alone	Living in refugee camps under difficult circumstances	
- Some of us don’t know English - Can’t get a job - Low level of skill	Low skills and poor employment outcomes	Post-migration contexts and realities
- Put your problems into alcohol - They tend to drink alcohol a lot - You don’t have a good quality of life	Impacts of poverty and unemployment	
- Coming to this new place - A new environment - Don’t want to operate with other people	Cultural differences	

**Table 3 ijerph-18-01534-t003:** Characteristics of face-to-face interview participants.

Participants	Gender	Age at Data Collection	Country of Origin	Year Arrived in Adelaide
1	Male	18	DRC	2010
2	Male	20	South Sudan	2008
3	Female	21	Liberia	2010
4	Male	23	South Sudan	2009
5	Male	20	South Sudan	2011
6	Male	23	Ethiopia	2003
7	Female	24	Liberia	2004
8	Male	20	Burundi	2007
9	Male	23	South Sudan	2007
10	Male	25	South Sudan	2007
11	Male	21	Liberia	2008
12	Male	22	DRC	2005
13	Female	25	South Sudan	2003
14	Female	23	South Sudan	2005
15	Male	20	South Sudan	2005
16	Female	25	South Sudan	2008
17	Female	21	Liberia	2008
18	Male	25	South Sudan	2006
19	Male	25	South Sudan	2003
20	Male	20	South Sudan	2001
21	Male	25	South Sudan	2006
22	Female	18	Liberia	2005
23	Female	23	South Sudan	2003

**Table 4 ijerph-18-01534-t004:** Characteristics of focus group youth participants.

Participants	Gender	Age at Data Collection	Country of Origin	Year Arrived in Adelaide
1	Male	25	South Sudan	2000
2	Female	20	Liberia	2005
3	Female	21	Somalia	2001
4	Female	23	Burundi	2005
5	Female	20	Liberia	2005
6	Female	23	Ghana	2004
7	Female	24	Ethiopia	2005

## Data Availability

The data used for the current study are not possible to share for ethical reasons. The Ethical review board (SBRECS) of Flinders University requires that the data are not shared without the consent of participants, who are currently not accessible.

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
