# Peer review of "Alcohol, Other Drugs Use and Mental Health among African Migrant Youths in South Australia"

_ijerph, 2021, doi:10.3390/ijerph18041534_

Round 1

Reviewer 1 Report

This paper presents the findings from a focus group and 23 individual interviews with young people who migrated to Australia from Africa, mostly as refugees. The authors focus on what these youth think about why other youth resort to the use of drugs.  There is very little that is new here.  One could say the same about just about any cohort of young people who resort to drug use due to social stressors. What we learn therefore is mainly stereotypes about drug use among others. The paper would have been more interesting if it had focused on how the interviewees learned to cope with the difficult experience of moving to a new country often without parents or other adult supports.

Author Response

The authors would like to say thank you to the reviewer for their suggestions made for improvement of the article around the research design and the conclusion. While these are important suggestions, it was not clear to the authors what part of the research design require improvement. However, we have read the methods section and provided more details in the text as much as possible.  We also like to let the reviewer know that our paper is novel and provides new information because: Our paper focuses on a specific setting and context of African refugee youths in South Australia.  To our knowledge and before writing this paper, we could not find any other authors who have conducted research in these populations, meaning that our paper if the first to provide the contextual information from these settings- the strength of the qualitative research of providing rich and specific information regarding participants. Similarly, we would add by letting the reviewer know that rather than seeing the paper as, “is mainly stereotypes about drug use among others”, we consider this paper to making an important contribution to understanding of contextual, the social and environmental insight that contributes to several issues including but not limited to drug use, alcohol, suicide and many others. While the suggestion that “the paper would have been more interesting if it had focused on how the interviewees learned to cope with the difficult experience of moving to a new country often without parent or adult supports” is a good recommendation, it was not the intention of this paper to focus on the coping strategies. The authors see this suggestion from the reviewer to focus on coping strategies as another potential paper from the study, for which we will make another submission as this aspect as well have some merits in it.

Reviewer 2 Report

The manuscript presents a study based on the qualitative analysis of 23 in-depth interviews with African immigrant youths in South Australia about substance abuse and mental health issues. The authors employed an acculturative stress framework to understand themes that emerged from these interviews. The interviews reveal numerous challenges that the youths have faced: loss of identity, transition to a new culture, unmet educational and recreational needs (especially for refugees); peer pressure, etc. The authors conclude that a unique combination of social and peer pressure, vulnerability, and deprivation are likely to cause mental health problems and substance use among the youths.

Overall, the study offers a fresh look into the lives of vulnerable youths that have had to traverse significant barriers in their lives. The manuscript is generally well-structured and well-written. The accounts given by the subjects well fit the narrative that the authors presented. I have one concern regarding the authors’ emphasis on acculturative stress. There can be other explanations for the incidence of substance abuse and mental health issues among the interviewees that, I believe, the authors should pay more attention to them in this paper. It is possible, for example, that the youths’ pre-migration experiences (e.g., the traumas of their past lives) made a more significant impact on their current behaviour than the barriers to their integration into the Australian society incurred in their post-migration lives. Nevertheless, I applaud the authors for their efforts (qualitative work is time-consuming and requires a personal touch) and look forward to their continued work in this area.

Author Response

The authors would like to thank this reviewer for his/her suggestion. The reviewer has raised a concern around the “authors’ emphasis on acculturative stress” and suggested a need to provide other explanations. To this end, we have added a sentence in the discussion to note that: While the paper emphasises the role of acculturative stress, it is important to acknowledge that there are also other factors that could contribute to AOD use and mental health issues among the participants. For example, pre-migration experiences such as traumas of past lives and losses could contribute to mental health issues among youths and use of AOD as coping mechanisms.

Reviewer 3 Report

This manuscript addresses a gap in the literature on alcohol and drug use and mental health among African migrants in Australia. The authors make a strong case for the need for this study and ground their approach in the acculturative stress model. 

I have a few suggestions to consider to strengthen the manuscript:

  1. Make a stronger connection between the social determinants of health and the conclusions in the discussion section. The authors argue for the study findings to be used in opportunities for folks providing mental health services, but the majority of study findings seem to relate not just to a lack of mental health services, but instead to a broader social environment that fosters distress, including lack of employment opportunities. 
  2. In the limitations, consider including how the study design of only including English speakers may limit the generalizability of study results. The fact that participants could speak English may be a factor in their resiliency but their experiences may not relate to other migrant youths who do not speak English. I think this is important to address. 
  3. In the discussion, connect the alcohol use to the quotes about risky behaviors and overall vulnerability of youth. 

More detailed comments and edits are included in the attached file. 

Author Response

The authors like to thank this reviewer for taking the time to provide editing and suggestions throughout the paper. The reviewer suggested that results section and the conclusion could be improved and provided three specific suggestions.

  • The authors have strenthened the conclusion section linking the conclusion to broader social environments that fosters distress.
  • The authors have added a sentence in the limitation section noting the transferability of the findings to youths who cannot speak English because their needs and barriers may be different from the identified needs of the youths studied.
  • We have added a sentence in the discussion section connecting alcohol use to the quotes about risky behaviour and overall vulnerability of youths.

Round 2

Reviewer 1 Report

I don't see any changes in this paper that make think it is helpful for understanding how migrant youth in Australia turn to drug use. The interviewees are not selected because of their knowledge of the problem and so we are left with mostly stereotypes about why youth turn to drugs. The abstract states that the findings show that "differences in cultural perspectives about AOD use that existed in Africa and Australia also shaped the experiences of social stressors." I don't see any evidence of this in the paper. The paper mainly says that youth who experience stressors, and they are probably severe, will turn to drugs. This is a stereotype that may or may not adequately describe the young migrants coming from Africa to Australia. By setting up the interviews to probe why their peers turn to drugs, the authors are only eliciting information that supports that premise. It would have been much better to ask how youth coming from Africa are coping with the challenges they face and explore those whether they involve drug use or not. In short, it is not clear that we are learning how youth coming from Africa to Australia are coping with the transition. All we are learning is the reasons some may turn to drugs, but these are just suppositions.